# A Concise Review of Scientific Research at EQ-SANS: Advancing Nanoscale Science Across Diverse Disciplines

## Abstract

The Extended Q-range Small-Angle Neutron Scattering (EQ-SANS) instrument at the Spallation Neutron Source (SNS), Oak Ridge National Laboratory (ORNL), has enabled investigations of nanoscale structures across diverse scientific disciplines. This review highlights key research advancements using EQ-SANS, spanning polymer science, biological and biomimetic systems, nanomaterials and colloids, energy materials, and environmental applications. We also discuss methodological developments and data analysis, including the integration of machine learning and artificial intelligence. Leveraging its unique capabilities, EQ-SANS has provided unprecedented insights into complex material behaviors, advancing both fundamental science and technological innovation.

## 1 Introduction: The EQ-SANS Instrument and Scope of this Review

The Extended Q-range Small-Angle Neutron Scattering (EQ-SANS) diffractometer at the Spallation Neutron Source (SNS) in Oak Ridge National Laboratory (ORNL) stands as a premier instrument for probing structural details across length scales ranging from approximately 0.5 nm to over hundreds of nm [1]. It offers wide neutron momentum transfer (Q) coverage, high intensity, and excellent wavelength resolution via time-of-flight and frame-skipping modes [2, 3]. Its ability to provide real-time, in-situ measurements, particularly for time-dependent phenomena and complex processes, distinguishes it as a powerful tool in materials science and beyond. The instrument features neutron optics optimized for transport and background minimization, and a two-dimensional $^3$He tube detector featuring high counting rates and efficiency [2]. Initial operations confirmed its design goals, marking a significant advancement in SANS instrumentation at pulsed spallation sources [3].

This review overviews diverse research at EQ-SANS, illustrating its impact on nanoscale science.. We aim to synthesize the key findings from a broad range of publications, organize them into thematic areas to showcase the instrument's versatility and the breadth of its scientific contributions. This review will cover advancements in polymer science and engineering, insights into biological and biomimetic systems, studies on nanomaterials and colloidal systems, research in energy materials and environmental applications, and significant methodological developments in data analysis, including the emerging role of artificial intelligence. By highlighting these achievements, we underscore EQ-SANS's crucial role in deciphering complex material structures and their relationship to mateirial properties, paving the way for future scientific discoveries and technological innovations.

## 2 Advancements in Polymer Science and Engineering

EQ-SANS has significantly contributed to understanding the intricate structures and behaviors of polymers, ranging from their self-assembly in solutions to their mechanical properties in bulk. The

Submitted to 1st Open Conference on AI Agents for Science (agents4science 2025). Do not distribute.

unique capabilities of SANS, particularly with deuterium labeling, allow for detailed insights into polymer conformation, phase behavior, and interactions with other components.

## 2.1 Polymer Conformation and Self-Assembly

Studies on bottlebrush polymers have clarified their structural evolution and scaling laws. Ahn et al. tracked PLA bottlebrushes during ROMP and observed elongated $\rightarrow$ globular $\rightarrow$ elongated transitions driven by excluded-volume effects [4]. Alaboalirat et al. established scaling relations linking structural parameters to backbone and side-chain degrees of polymerization [5]. Atomistic side-chain conformations have also been resolved with combined SANS and MD [6].

Block-copolymer self-assembly in water has been mapped extensively. Do et al. combined mesoscale simulations and SANS to identify spherical micellar, lamellar, and reverse-micellar phases of Pluronic L62 [7]. Jang et al. showed Pluronic blends form temperature-sensitive unilamellar vesicles with tunable size and bilayer thickness [8]. Additives modulate micellization: nucleoside analogues reduce L62 micelle size and enhance core hydration [9], while ionic liquids depress transition temperatures and favor larger aggregates [10].

Guest–host architecture and hydration have been probed by contrast variation. In PAMAM dendrimers, surfactants localize to the periphery, inducing steric crowding and reduced hydration [11]. Amphiphilic invertible polymers form cylindrical core–shell micelles that invert in toluene [12]. Isotopic-label SANS further shows star-polymer branches fold inward via solvation effects [13, 14].

## 2.2 Polymer Mechanics and Dynamics

The mechanical properties and dynamics of polymers and polymer composites have been extensively studied. The impact of backbone rigidity on the thermomechanical properties of semiconducting polymers with conjugation break spacers was quantified, revealing that increased spacer length enhances flexibility and reduces elastic modulus [15]. The chain stiffness of donor-acceptor conjugated polymers in solution was measured by SANS, showing that side-chain size and branching significantly influence persistence lengths, correlating with charge-carrier mobility [16]. Investigations into poly(3-alkylthiophenes) (P3ATs) have demonstrated that side chain isomerism influences their rigidity, with branched side chains promoting greater flexibility [17, 18, 19].

The impact of polydispersity on microphase separation in thin films of lamellar-forming diblock copolymers has been explored, demonstrating that increasing polydispersity reduces the number of lamellar strata and induces conformational asymmetry [20]. The phenomenon of structural anisotropy relaxation in deformed polymers has been quantitatively investigated, revealing a scaling law where the relaxation rate is proportional to Q at high Q and short times, challenging classical Rouse and tube models [21, 22]. Furthermore, a generalized Zimm plot approach has been introduced to quantify molecular deformation in polymer melts using SANS, providing a model-independent analysis of spatially dependent molecular deformation [23].

Structural information from SANS is often used to complement other experimental techniques, such as neutron spin-echo (NSE) or rheometry, for a more complete understanding of polymer dynamics. Studies on associative polymer networks have shown that sticker clustering increases relaxation times, attributed to cooperative dissociation of multiple bonds, while also surprisingly accelerating diffusion due to loop defects [24]. Hindered segmental dynamics in associative protein hydrogels due to transient binding have also been quantified using NSE [25]. The dynamics of Li+ transport in poly(ethylene oxide) (PEO) based electrolytes have been investigated using neutron spin-echo (NSE), dielectric spectroscopy, and MD simulations, revealing a strong coupling between dc conductivity and dielectric $\alpha$ relaxation time [26]

## 2.3 Responsive Polymer Systems

Temperature-responsive polymer systems have been a key area of research. Hyatt et al. investigated poly(N-isopropylacrylamide) (pNIPAM) microgels, observing mass segregation at the particle periphery and a decrease in the polymer network length scale at high temperatures, linked to charge segregation [27, 28, 29]. The self-assembly of thermo-reversible block copolymers coating single-walled carbon nanotubes has been characterized, showing tunable encapsulation structures [30, 31]. The phase behavior of Pluronic P65 blended with 5-methylsalicylic acid (5mS) exhibited a closed

loop-like phase behavior, transitioning from isotropic to ordered and back to isotropic with increasing temperature [32]. Additionally, temperature-responsive polymersomes composed of poly(3- methy-n-vinylcaprolactam)-block-(poly(n-vinylpyrrolidone) diblock copolymers have been synthesized for reduced doxorubicin-induced cardiotoxicity [33].

The study of water-soluble polymers across multiple concentration regimes has quantified the number of hydration water molecules associated with different polymers using contrast-variation SANS, leading to improved understanding of water-polymer interactions [34, 35, 36, 37]. The self-assembly of a multifunctional ionic block copolymer in selective solvents has been elucidated, forming ellipsoidal core-shell micelles with varying sizes and aggregation numbers depending on concentration [38]. The dynamic implications of noncovalent interactions in amphiphilic single-chain polymer nanoparticles (SCNPs) have also been explored, demonstrating how these interactions restrict internal relaxations and guide the design of biomimetic materials [39].

# 3 Insights into Biological and Biomimetic Systems

EQ-SANS has been instrumental in unraveling the complex structures and dynamics of biological and biomimetic systems, providing a deeper understanding of fundamental biological processes and informing the design of advanced biomaterials.

## 3.1 Protein Structure and Dynamics

The molecular conformation and binding activity of crucial proteins, such as the tumor suppressor NF2/Merlin, have been investigated, revealing a rheostat model of function where conformation and binding are not simply open or closed states [40]. The dynamic structure of the scaffolding protein NHERF1, and how disease-associated point mutations alter its flexibility and signaling complex assembly, has been characterized using NMR and SANS [41]. The structural information of a-Catenin obtained from EQ-SANS helped understanding nanoscale dynamics too, both in solution and in complex with F-actin, suggesting its dynamic conformations enable mechanosensing [42]. Phospho-mimetic mutation of the multi-domain scaffolding protein NHERF1 and buffer salt concentration was show to influence the protein's nanoscale dynamics and binding kinetics [43].

The conformational behavior of intrinsically disordered proteins (IDPs) under macromolecular crowding conditions has been explored, revealing a biphasic response of compaction followed by expansion for FlgM [44]. The folding propensity of IDPs by osmotic stress has been investigated, highlighting the importance of hydration changes in IDP folding [45]. The solution structures of NADPH-dependent assimilatory Sulfite Reductase (SiR) have been modeled, providing insights into electron transfer mechanisms and conformational changes upon subunit binding and changes in redox state [46, 47, 48]. Furthermore, the structural ensemble of an IDP complex (NCBD/ACTR complex, that is associated with breast and ovarian cancers) has been characterized using an integrated approach combining residue-specific deuterium labeling SANS, MD simulations, and deep learning algorithms [49].

## 3.2 Membrane Biophysics

EQ-SANS has clarified the structure and dynamics of lipid bilayers, often in the presence of peptides and other biomolecules. PIP2 clusters the cell-adhesion molecule CD44 and mediates assembly of CD44–Ezrin heterocomplexes, while the conformation of Ezrin bound to PIP2 and F-actin illuminates the membrane–cytoskeleton interface [50, 51].

Cholesterol may promote protein binding by altering membrane electrostatics and solvation [52]. Joint SANS/SAXS resolved the molecular structure of sphingomyelin in fluid bilayers, informing lipid packing and reconciling differences between NMR- and scattering-derived parameters [53].

Peptide–membrane interactions strongly remodel bilayer structure. An HIV-1 gp41 fusion-peptide derivative undergoes a helix-to-sheet transition that induces localized negative curvature and increased rigidity, changes consistent with fusion promotion [54, 55, 56]. Alamethicin disrupts cholesterol distribution and homogenizes laterally heterogeneous phases [57], while melittin causes concentration-dependent thickening or thinning linked to lipid redistribution [58].

Lipid transport is sensitive to environment and architecture: methanol accelerates DMPC flip-flop and intervesicle transfer [59]; bicelles exhibit faster transfer than vesicles, attributed to interfacial defects from hydrophobic mismatch [60, 61].

Lipid domains (rafts) have been probed at nanoscopic scales. The bending modulus of domains was isolated, showing modulus mismatch drives lateral heterogeneity [62]. In vivo evidence for domains in *Bacillus subtilis* membranes was obtained using isotopic labeling [63, 64, 65]. Rafts appear to buffer membrane physical properties, stabilizing diffusion and bending modulus with temperature changes [66]. Ergosterol shows nonstereotypical distributions and concentration-dependent rigidifying/softening effects, and promotes jump diffusion [67].

### 3.3 Biomaterials, Bio-Inspired Systems, and Drug Delivery

EQ-SANS studies have advanced the design of biomaterials for drug delivery and other applications. Lignin-graft-poly(lactic-co-glycolic acid) biopolymers have been synthesized for polymeric nanoparticle synthesis, exhibiting a core-shell structure and showing potential as a delivery system [68]. Recombinant globular fusion proteins have been engineered to self-assemble into vesicles with tunable size and membrane structure, with EQ-SANS measurements quantifying membrane thickness and confirming temperature-dependent transitions critical for designing protein-based delivery systems. [69]. PEGylation site-dependent structural heterogeneity of monoPEGylated human parathyroid hormone fragment hPTH(1-34) has been investigated, showing core-shell cylindrical structures with size variations potentially impacting pharmacokinetics [70]. The spontaneous nanostructures of bicellar mixtures and the effects of temperature, salinity, concentration, and PEGylated lipids on nanodisc-to-vesicle transitions have been characterized, revealing nanodisc stabilization by PEGylation [71, 72]. Nucleopore-inspired polymer hydrogels for selective biomolecular transport have been developed, demonstrating selective permeability based on binding interactions between biomolecules and the hydrogel [73].

The structural and dynamic heterogeneity in associative networks formed by artificial coiled-coil proteins has been explored, revealing various static length scales and superdiffusive regimes [74, 75]. Alginate/PEO-PPO-PEO composite hydrogels with thermally-active plasticity have been developed, demonstrating increased elastic modulus and fracture stress above the lower gelation temperature [76]. The enhancement of polymer thermoresponsiveness and drug delivery across biological barriers by adding small molecules to poloxamer has also been demonstrated [77]. The assembly of lipid-hyaluronan complexes in osteoarthritic conditions, and the influence of HA concentration and molecular weight on their structure, has been investigated, with implications for cartilage lubrication [78].

### 3.4 Plant Biology

Research at EQ-SANS has also contributed to plant biology and bio-inspired materials. The structural changes of the CESA1 catalytic domain of Arabidopsis cellulose synthesis complex provided evidence for CESA trimers, supporting the "hexamer of trimers" model for cellulose synthesis [79]. Dynamic in vivo monitoring of granum structural changes in Ctenanthe setosa during drought stress and recovery has revealed rapid recovery of granum structure upon rewatering, preceding functional and biochemical recovery [80]. The functional in vitro diversity of an intrinsically disordered plant protein (COR15A) during freeze-thawing, encoded by its structural plasticity, has been investigated [81]. Evidence for lignin-carbohydrate complexes from studies of transgenic and wild type switchgrass and a model lignin-pectin composite has been provided, suggesting their role in biomass recalcitrance [82].

## 4   Nanomaterials and Colloidal Systems

EQ-SANS has been a crucial tool for characterizing the structure and behavior of diverse nanomaterials and colloidal systems, from metal nanoparticles to complex hierarchical assemblies.

## 4.1 Nanoparticle Synthesis and Characterization

Gold nanoparticle (AuNP) architectures span from 2D superlattices formed within polymer vesicle layers via hydrophobic interactions—useful for traceable nanoreactors and electron-exchange platforms [83]—to binary AuNP/Brij 58/water superlattices whose structures are AuNP-size dependent and thermally responsive [84]. Charge-tunable surfactant capping also yields water-redispersible, highly stable AuNPs suitable for biomedical processing [85].

Pluronic triblock copolymers markedly improve BNNT dispersibility in water [86]. BNNTs further self-assemble into 2D hexagonal arrays in block-copolymer matrices (with piezoelectric potential) [87] and into highly ordered 2D binary superlattices with cationic surfactant vesicles via electrostatics [88].

Silica–conjugated polymer hybrid fluorescent nanoparticles prepared by surface-initiated polymerization exhibit tunable optical responses [89]. Core–shell nanospheres with smectic hydrophobic cores and PEG shells show concentration-dependent structures relevant to drug release [90].

## 4.2 Colloidal Interactions and Dynamics

The densification of ionic liquid molecules within hierarchical nanoporous carbon structures has been revealed, showing significantly higher room temperature ionic liquid (RTIL) densities compared to the bulk fluid due to strong affinity between the RTIL cation and the carbon surface [91]. A dense microemulsion system formed with an ionic liquid has been studied, revealing a two-phase system of water-in-oil and bicontinuous microemulsions [92]. The internal structure of polyelectrolyte complex coacervates has been comprehensively evaluated, determining chain dimensions, validating sticky reptation theory, and quantifying salt doping effects on dynamics [93].

The multiscale structure of asphaltenes in various solvents has been investigated, showing that asphaltene clusters persist to dilute concentrations and follow a fractal scaling law [94]. The aggregation behavior of high-purity vanadyl petroporphyrins (VOPPs) and their impact on asphaltene aggregation have been explored, with VOPPs forming small nanoaggregates and influencing asphaltene self-assembly [95]. The interfacial behavior of purified VOPPs and their influence on asphaltene film formation at the water-oil interface has also been studied, revealing that VOPPs can form monolayers with low tension but do not prevent thick asphaltene films [96].

The effect of magnetization on the gel structure and protein electrophoresis in polyacrylamide hydrogel nanocomposites has been investigated, showing morphological changes and reduced pore size correlating with protein separation performance [97]. An interface-driven stiffening mechanism in polymer nanocomposites has been identified, where chains desorb from nanoparticle surfaces and entangle with free chains during resting periods, leading to interfacial hardening [98]. The synergistic role of temperature and salinity in the aggregation of nonionic surfactant-coated silica nanoparticles has been demonstrated, promoting surfactant adsorption and silica aggregation [99].

## 4.3 Advanced Nanostructure Fabrication

The formation of uniformly aligned chiral photonic films from cellulose nanocrystals (CNCs) within a thin capillary has been demonstrated, accelerating the ordering process and leading to highly oriented films [100]. Multicompartmental microcapsules from star copolymer micelles have been fabricated using layer-by-layer assembly, possessing nanoporous shells capable of storing different components [101]. The structural study of star polyelectrolytes and their porous multilayer assembly in solution revealed contraction of cationic star polyelectrolyte arms and disruption of spatial ordering upon salt addition [102].

A novel bio-templating method for synthesizing chiral metal-organic frameworks (MOFs) from achiral precursors using chiral nematic nanocelluloses has been developed, resulting in chiral zeolitic imidazolate frameworks (ZIFs) with enantioselective sensing abilities [103]. The kinetically controlled assembly of conjugated polymer (CP) nanostructures has been investigated, yielding hierarchically organized CP systems with distinct optoelectronic properties through in situ polymerization [104]. The control of molecular ordering in water-soluble conjugated polymers through thermally-controlled and surfactant-guided assembly has also been shown to influence electronic interaction and optical function [105]. The discovery of iridescence in nematic liquid crystals composed of

nanoplates, even without long-range periodicity, has opened new possibilities for photonic materials [106].

# 5 Energy Materials and Environmental Applications

EQ-SANS has been a valuable tool for understanding the structure and dynamics of materials relevant to energy storage, conversion, and environmental remediation.

## 5.1 Battery and Energy Storage Materials

In-situ observation of solid electrolyte interphase (SEI) formation in ordered mesoporous hard carbon has provided real-time information on the composition and microstructure of electrodes in lithium half-cells [107]. The framework expansion of ordered mesoporous hard carbon anodes with ionic-liquid electrolytes has been observed, highlighting the importance of framework expansion and SEI formation for stable cycling [108]. Insight into SEI formation in bis(fluorosulfonyl)imide based ionic liquid electrolytes has been gained, confirming the protective role of the bis(fluorosulfonyl)imide (FSI-) anion against 1-ethyl-3-methylimidazolium (EMIm) cation co-intercalation [109].

Structural investigation using EQ-SANS has contributed to the understanding of the solution dynamics and binding of polyvinylidene fluoride (PVDF) binder with silicon, graphite, and Nickel Manganese Cobalt (NMC) materials have been investigated, revealing incomplete binder adsorption on silicon, disrupting percolation pathways and leading to poor cycling performance [110]. The origin of rate limitations in solid-state polymer batteries from constrained segmental dynamics within the cathode has been identified, where PEO chains adsorb onto lithium iron phosphase (LFP) particles, reducing Li+ mobility [111]. The structural properties of quaternary ammonium-based ionic liquids have been studied, characterizing short- and long-range liquid structure indicative of alternating polarity, charge, and neighboring domains [112].

The effect of metal ion intercalation on the structure of MXene and water dynamics on its internal surfaces has been explored, showing that K+ intercalation enhances structural homogeneity and water stability in MXenes [113]. The structure-performance relationships of lithium-ion battery cathodes have been revealed by contrast-variation SANS, deconvoluting carbon and binder phases and correlating solvent-accessible carbon black surface area with diminished capacity retention [114].

## 5.2 Catalysis and Adsorption

The linking of $CO_2$ sorption performance to polymer morphology in aminopolymer/silica composites has been achieved through neutron scattering, revealing that poly(ethylenimine) (PEI) forms a thin conformal coating on pore walls, with additional polymer aggregating into plugs [115]. The interactions of an imine polymer with nanoporous silica and carbon in hybrid adsorbents for carbon capture have been investigated, showing strong densification of PEI in carbon nanopores and its impact on capture capacity [116]. The distribution and mobility of PEI within mesoporous silica after multiple $CO_2$ sorption-regeneration cycles have been probed, highlighting the crucial role of water in maintaining PEI distribution and mobility [117]. The underlying roles of polyol additives in promoting $CO_2$ capture in PEI/silica adsorbents have been elucidated, showing that poly(ethylene glycol) (PEG) displaces wall-bound PEI, making amines more accessible for $CO_2$ sorption [118].

The adsorption and catalytic activity of gold nanoparticles in mesoporous silica have been studied, demonstrating that confined gold nanoparticles (AuNPs) can withstand aggregation under high salinity, retaining catalytic activity [119]. Characterization of nano-assemblies inside mesopores using neutron scattering has extended a method to include interparticle correlations, enabling qualitative characterization of surfactants and nanoparticles adsorbed in cylindrical pores [120].

## 5.3 Environmental Remediation and Sustainable Materials

Research into solvent extraction systems for heavy metal ions has utilized SANS. The microscopic structures of tri-n-butyl phosphate (TBP)/n-octane mixtures have been investigated, revealing that TBP self-associates into ellipsoidal assemblies [121]. EQ-SANS data have provided complementary information to the neutron polarization analysis to accurately determine coherent scattering intensity from biphasic solvent extraction systems, crucial for structural analysis of extracted complexes [122].

A telescoping view of solute architectures in a complex fluid system involved in metal refining and purification has elucidated the hierarchical aggregation of metal-ligand complexes [123]. Proton chelating ligands have been shown to drive improved chemical separations for rhodium, with SANS characterizing the outer-sphere assembly of the Rh(III) complex [124].

The nanoscopic structure of borosilicate glass with additives for nuclear waste vitrification has been investigated, revealing the impact of additives on microphase separation and void formation [125]. The structure and water-binding in Alkali-Silica Reaction (ASR) sol and gel have been studied, showing how alkali cation type influences agglomerate structures and water binding ability, with implications for concrete durability [126]. The impact of fuel on surfactant microstructure of firefighting foam has been investigated, providing insights into the factors controlling firefighting performance and aiding in the development of environmentally friendly foams [127].

## 5.4 Organic Photovoltaics and Flexible Electronics

The role of additives in improving the performance of bulk heterojunction organic solar cells has been investigated, revealing that additives induce a shift in morphology from solution to film, leading to hierarchical structures with optimum crystallinity [128, 129]. The morphology of active layers in all-polymer photovoltaic cells has been characterized, showing P3HT crystallites dispersed within an amorphous matrix, with graphene addition affecting electronic properties but not film structure [130]. The critical role of electron-donating thiophene groups on the mechanical and thermal properties of donor-acceptor semiconducting polymers has been elucidated, showing their anti-plasticizing effect and providing design rules for stretchable electronics [131]. The concept of disorder-tolerant semiconducting polymers has been approached through computer-aided molecular design, identifying pyrazine and difluorothiophene combinations for high torsional barrier and planarity, leading to efficient n-doping and high electrical conductivities [132].

# 6 Methodological Developments and Data Analysis at EQ-SANS

Beyond its direct scientific applications, EQ-SANS has been a hub for significant advancements in neutron scattering methodologies and data analysis, particularly integrating computational techniques and machine learning.

## 6.1 Advancements in AI and Machine Learning for SANS Data Analysis

The application of deep learning-based super-resolution techniques has been explored to accelerate SANS data collection. Chang et al. demonstrated the feasibility of reconstructing high-resolution scattering data from low-resolution inputs using a deep convolutional neural network, potentially speeding up experimental workflows [133]. A machine learning (ML) inversion scheme has been introduced for determining the effective interaction in colloids directly from scattering data, offering superior accuracy and efficiency compared to traditional parametric methods [134, 135].

Deep learning has also been leveraged to decipher the scattering of mechanically driven polymers. Ding et al. presented a Variational Autoencoder (VAE) approach to analyze two-dimensional scattering data of semiflexible polymers under external forces, enabling significantly faster extraction of polymer parameters compared to traditional fitting procedures [136]. An integration of machine learning with Monte Carlo simulations has been developed to model kinked CANAL ladder polymer structures, uncovering features conventional methods fail to capture [137].

Model-free approaches for profiling polydisperse soft matter using small angle scattering have been developed. Huang et al. introduced a strategy that uses moment expansion to extract central moments and reconstruct the size distribution function without bias, validating the approach on L64 Pluronic micelles [138]. A novel method for reconstructing the neutron scattering length density profile from SANS intensity profiles has been presented, utilizing a universal operator and PhaseLift framework to eliminate the need for predefined models and mitigate error propagation [139]. Bayesian statistical inference using Gaussian Process Regression (GPR) has also been explored to reconstruct high-fidelity scattering data from sparse SANS measurements, maximizing experimental efficiency and enabling high-throughput studies [140].

## 6.2 Probing Deformed Systems: Understanding Structure under Flow and Stress

EQ-SANS has facilitated the study of materials under various mechanical stresses, providing insights into their structural response. A portable hydro-thermo-mechanical loading cell has been developed for in-situ SANS studies of proton exchange membranes, allowing for tensile loading of samples immersed in liquid environments at controlled temperatures [141]. This cell has been used to investigate the mechanical properties and microstructure changes of Nafion membranes under immersed conditions, revealing a disorder-order transition with increasing temperature and water uptake [142].

The influence of elongation-induced concentration fluctuations on segmental friction in polymer blends has been investigated using rheology and SANS, demonstrating that viscoelastic asymmetry leads to demixing and apparent friction enhancement [143]. The local elasticity in nonlinear rheology of interacting colloidal glasses has been revealed by in-situ SANS and rheological measurements, identifying a transient elasticity zone (TEZ) at the particle level that governs shear-thinning behavior [144, 145]. An exact inversion method for extracting orientation ordering from small-angle scattering has been introduced, accurately determining the orientation distribution function (ODF) of sheared interacting rods [146, 147]. Furthermore, the potentials of SANS for understanding the structure-property relation of 3D-printed materials have been explored, correlating microstructure of carbon fiber-embedded composites with mechanical strength and highlighting the impact of carbon fiber on polymer chain conformation and interfacial structure [148].

## 6.3 Other Instrument Performance and Enhancements

Significant efforts have been made to improve the accuracy and efficiency of SANS data acquisition and processing at ORNL. Corrections for the geometric distortion of the tube detectors on SANS instruments have been developed, improving data quality [149, 150]. The data processing scheme for the EQ-SANS diffractometer has been refined to be fast, versatile, and highly automated, directly converting event files into scattering intensity data and enabling time-slicing for time-resolved experiments [151].

The phenomenon of inelastically scattered neutrons from water on a time-of-flight SANS instrument has been investigated, revealing a significant inelastic process where scattered neutrons exhibit energies consistent with room-temperature thermal energies, emphasizing the need for careful data processing for hydrogenous materials [152].

A unified user-friendly instrument control and data acquisition system (IC-DAS) has been developed for the ORNL SANS instrument suite, improving ease of use and efficiency for researchers conducting SANS experiments [153]. Furthermore, the EQ-SANS Assisting Chatbot (ESAC) has been introduced, leveraging Large Language Models (LLM) and Retrieval-Augmented Generation (RAG) to enhance user experience by providing an interactive reference and automating script generation [154].

# 7 Conclusion and Future Perspectives

The extensive research conducted using the EQ-SANS instrument has significantly impacted various scientific fields. In polymer science, it has provided key insights into polymer structures and properties, while in biology, it has helped to clarify complex protein and membrane dynamics. EQ-SANS has also been crucial for advancing the understanding of nanomaterials and colloids and has played a vital role in research on energy materials, such as batteries, and sustainable technologies. The instrument's versatility is highlighted by the more than 300 publications it has contributed to, underscoring its central role in the global scientific community.

The future of EQ-SANS is promising, with planned upgrades to the accelerator and the consequent increased neutron flux expected to enhance its capabilities significantly. These improvements will allow scientists to conduct more complex experiments on smaller samples, observe faster changes in materials, and resolve even finer structural details. This will continue to push the frontiers of nanoscale science and drive innovation in crucial areas. The ongoing collaboration between advanced instrumentation and interdisciplinary research will ensure that EQ-SANS remains a leading facility for scientific discovery for many years.

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

## A    Technical Appendices and Supplementary Material

Technical appendices with additional results, figures, graphs and proofs may be submitted with the paper submission before the full submission deadline, or as a separate PDF in the ZIP file below before the supplementary material deadline. There is no page limit for the technical appendices.


