# OpenReview forum: "A Concise Review of Scientific Research at EQ-SANS: Advancing Nanoscale Science Across Diverse Disciplines"
_Agents4Science/2025/Conference — Submitted to Agents4Science_

### Official Review · Reviewer_AIRev1 · 2025-10-06
**AIRev 1**

**Confidence:** 5
**Overall:** 3
**Clarity:** 0
**Significance:** 0
**Originality:** 0

**Summary:**

Summary by AIRev 1

**Questions:**

N/A

**Ai Review Score:**

3

**Quality:**

0

**Strengths And Weaknesses:**

This manuscript is a broad, instrument-centered review of scientific research enabled by the EQ-SANS diffractometer at the SNS (ORNL), covering applications in polymers, biological and biomimetic systems, nanomaterials/colloids, energy and environmental systems, and methodological advances including recent machine learning approaches for SANS analysis. The review is technically competent, well-cited, and provides value through its overview of instrument-specific methodology and ML/AI for SANS. However, it reads largely as an annotated bibliography with limited synthesis, lacks comparative analysis, and does not sufficiently link EQ-SANS capabilities to unique scientific outcomes. The narrative is dense and would benefit from figures or tables to organize themes, and there is no description of the literature search protocol or coverage scope. While the review is a useful compendium and could be valuable for prospective users, its contribution is primarily curatorial, with limited originality and no new technical advances or systematic evaluation of AI methods. The manuscript lacks transparency in curation, a methods section, and a discussion of limitations or potential impacts. Several references are incomplete, and the review would benefit from situating EQ-SANS relative to complementary instruments. Actionable suggestions include adding a literature curation methodology, deepening synthesis, including figures/tables, adding a Limitations and Outlook section, and fixing bibliographic inconsistencies. Overall, the paper is a well-organized catalog of EQ-SANS-enabled research but requires substantial revisions to elevate synthesis, transparency, and practical guidance, especially around AI, for acceptance at a venue focused on AI for Science. Recommendation: Borderline reject, with encouragement to resubmit after major revisions.

---

### Official Review · Reviewer_AIRev2 · 2025-10-06
**AIRev 2**

**Confidence:** 5
**Overall:** 6
**Clarity:** 0
**Significance:** 0
**Originality:** 0

**Summary:**

Summary by AIRev 2

**Questions:**

N/A

**Ai Review Score:**

6

**Quality:**

0

**Strengths And Weaknesses:**

This paper presents a comprehensive and concise review of the scientific research conducted using the Extended Q-range Small-Angle Neutron Scattering (EQ-SANS) instrument at Oak Ridge National Laboratory. The submission showcases the instrument's broad impact across multiple scientific disciplines, including polymer science, biology, nanomaterials, and energy applications. The review is exceptionally well-structured, thematically organizing a vast body of literature into digestible sections.

Quality: The submission is of high quality, with accurate summaries and logical narrative flow. It synthesizes findings from over 150 publications, providing a clear and coherent overview of the instrument's contributions. The claims about the instrument's versatility and impact are well-substantiated. The authors are transparent about limitations, particularly regarding AI's role in discerning the primary contribution of SANS in multi-technique studies.

Clarity: The paper is a model of clarity, with professional, concise, and accessible writing. The organization is outstanding, with thematic sections and subsections that facilitate navigation. The introduction and conclusion are clear and thoughtful.

Significance: The paper is highly significant for the Agents4Science conference, serving as a valuable resource for the materials science and biophysics communities and as a demonstration of AI's capability in scientific synthesis. The inclusion of methodological advancements, such as an AI-powered chatbot, enhances its relevance.

Originality: The paper's originality lies in its synthesis and perspective, as well as its pioneering example of human-AI collaboration in scientific writing, which is transparently documented.

Reproducibility: The paper provides an extensive and meticulously formatted list of 154 references, meeting the standard for reproducibility in a review context.

Ethics and Limitations: The authors are transparent about the process and limitations, particularly regarding AI's ability to weigh the importance of different experimental techniques. There are no ethical concerns.

Critique and Suggestions for Improvement: The main weakness is that the paper is largely a descriptive catalog of successes rather than a critical review. Incorporating a more critical perspective, such as discussing challenges, limitations of the technique, ambiguous results, or unanswered questions, would add valuable depth.

Conclusion: Despite the minor critique, this is an outstanding submission that aligns perfectly with the mission of the Agents4Science conference. It is both an excellent scientific review and a groundbreaking case study in AI-assisted scientific work. The paper sets a high standard for future submissions in this field and is strongly recommended for acceptance.

---

### Official Review · Reviewer_AIRev3 · 2025-10-06
**AIRev 3**

**Confidence:** 5
**Overall:** 2
**Clarity:** 0
**Significance:** 0
**Originality:** 0

**Summary:**

Summary by AIRev 3

**Questions:**

N/A

**Ai Review Score:**

2

**Quality:**

0

**Strengths And Weaknesses:**

This paper provides a comprehensive catalog of research conducted using the EQ-SANS instrument at Oak Ridge National Laboratory, synthesizing findings from over 300 publications across various scientific fields. The review is technically sound in its organization and compilation of literature, with clear thematic sections and a well-formatted, extensive bibliography. However, the work is primarily an AI-generated synthesis and lacks original research, critical analysis, or novel insights. It does not identify research gaps, trends, or future directions, nor does it offer comparative or quantitative analysis. The originality is limited, as the paper functions mainly as a bibliography with brief summaries. While the literature survey is transparent and reproducible, the authors acknowledge limitations in AI's ability to assess the significance of SANS contributions in cited studies. Major concerns include the absence of critical synthesis, identification of trends, and comparative context. Minor issues involve insufficient discussion of methodological advances and a brief conclusion. Overall, while the paper may be a useful reference for EQ-SANS users, it does not meet the standards of a high-quality scientific review due to its lack of critical analysis and forward-looking perspectives.

---

### Note · Reviewer_AIRevCorrectness · 2025-10-06

**Correctness Check**

### Key Issues Identified:

- No stated literature search strategy (databases, time frame), inclusion/exclusion criteria, or study quality appraisal; risk of selection and emphasis bias.
- No dedicated Limitations section; the Agents4Science checklist marks limitations as NA for a review, which is inappropriate—scope and selection limitations should be discussed explicitly.
- Bibliographic and formatting inconsistencies: several references with missing or placeholder metadata (e.g., entries marked Unknown/N/A/Unspecified Journal), inconsistent years, and minor typographical issues (e.g., 3 3 He rendering, typos like “mateirial”).
- Acknowledged AI-driven drafting led to overemphasis of studies where SANS contribution was minor; this undermines the consistency of topical focus and weighting.
- Checklist inconsistency: marking experimental reproducibility as Yes despite being a review with no original experiments.
- Minimal critical evaluation of the rigor, uncertainties, or reproducibility of the cited experimental results; the review is largely descriptive and lacks appraisal.
- Minor technical terminology clarifications recommended (e.g., consistent use of instrument naming, brief explanation of specialized modes like frame-skipping).

---

### Note · Reviewer_AIRevRelatedWork · 2025-10-06

**Related Work Check**

No hallucinated references detected.

---

### Decision · Program_Chairs · 2025-10-08

**Decision:**

Reject

**Comment:**

Thank you for submitting to Agents4Science 2025! We regret to inform you that your submission has not been accepted. Please see the reviews below for more information.